



Natural Hazards
and Earth System
Sciences

# Delimitation of flood areas based on a calibrated a DEM and geoprocessing: case study on the Uruguay River, Itaqui, southern Brazil

**Paulo Victor N. Araújo[1,2], Venerando E. Amaro[1], Robert M. Silva[3], and Alexandre B. Lopes[4]**

[1]Postgraduate Program in Geodynamics and Geophysics (PPGG), Department of Geology,
Federal University of Rio Grande do Norte, P.O. Box 1524, Natal-RN, 59078-970, Brazil
[2]Federal Institute of Education, Science and Technology of Rio Grande do Norte, Macau-RN, 59500-000, Brazil
[3]Federal University of Pampa, Itaqui-RS, 96650-000, Brazil
[4]Center of Sea Studies, Federal University of Parana, P.O. Box 61, Curitiba-PR CE1, 83255-976, Brazil

**Correspondence:** Paulo Victor N. Araújo (paulo.araujo@ifrn.edu.br)

Received: 14 July 2018 – Discussion started: 15 August 2018
Accepted: 2 January 2019 – Published:

**Abstract.** TS1 TS2 Flooding is a natural disaster which affects thousands of riverside, coastal, and urban communities causing severe damage. River flood mapping is the process of determining inundation extents and depth by comparing historical river water levels with ground surface elevation references. This paper aims to map flood hazard areas under the influence of the Uruguay River, Itaqui (southern Brazil), using a calibration digital elevation model (DEM), historic river level data and geoprocessing techniques. The temporal series of maximum annual level records of the Uruguay River, for the years 1942 to 2017, were linked to the Brazilian Geodetic System using geometric leveling and submitted for descriptive statistical analysis and probability. The DEM was calibrated with ground control points (GCPs) of high vertical accuracy based on post-processed high-precision Global Navigation Satellite System surveys. Using the temporal series statistical analysis results, the spatialization of flood hazard classes on the calibrated DEM was assessed and validated. Finally, the modeling of the simulated flood level was visually compared against the flood area on the satellite image, which were both registered on the same date. The free DEM calibration model indicated high correspondence with GCPs ($R^2 = 0.81$; $p < 0.001$). The calibrated DEM showed a 68.15 % improvement in vertical accuracy (RMSE = 1.00 m). Five classes of flood hazards were determined: extremely high flood hazard, high flood hazard, moderate flood hazard, low flood hazard, and non-floodable. The flood episodes, with a return time of 100 years, were modeled with a 57.24 m altimetric level. Altimetric levels above 51.66 m have a high potential of causing damage, mainly affecting properties and public facilities in the city's northern and western peripheries. Assessment of the areas that can potentially be flooded can help to reduce the negative impact of flood events by supporting the process of land use planning in areas exposed to flood hazard.

## 1 Introduction

Flooding, as a major natural disaster, affects many parts of the world including developed countries, and has severe impacts on populations and causes socioeconomic damage. Due to this kind of natural disaster, billions of dollars in infrastructure and property damage, as well as hundreds of human lives, are lost each year (Demir and Kisi, 2016; Elnazer et al., 2017). It is understood that flood risks and hazards will not subside in the future and, with the onset of climate change, flood intensity and frequency will threaten many regions of the world (IPCC, 2014).

These hazards can be prevented and reduced by providing reliable information to the public about the flood hazard through flood inundation maps (Alaghmand et al., 2010; Demir, 2015 TS3). This information can, for example, assist urban management or even help the rescue operations during these events (Cook and Merwade, 2009), thus helping the

communities directly to avoid or mitigate such future losses and damage (Arrighi et al., 2013; Savage et al., 2014; Speckhann et al., 2017).

Flood hazard maps therefore need to be created as they provide a basis for the development of flood risk management plans. These plans need to be effectively communicated to various target groups, including decision makers, emergency response units and the public, as a measure to reduce flood risk by integrating different interests, and potential conflicts over space and land use in a city (Ouma and Tateishi, 2014). However, creating flood maps is a complex process which is affected by the input data, flow design, and consistent topographic information (APFM, 2013). A major aid in the construction of flood maps is use of geoprocessing (Chen et al., 2009; Sarhadi et al., 2012; Demir and Kisi, 2016; Ovando et al., 2016; Liu and Yamazaki, 2018). Geoprocessing is a set of techniques based on the study of spatially distributed information in order to describe the characteristics of the phenomenon under investigation in the whole area of interest (Costa and Lourenço, 2011). The construction process of flood maps requires an understanding of flow dynamics over the floodplain, topographic relationships and sound judgments of the modeler (Noman et al., 2001; Sinnakaudan et al., 2003; Sanders, 2007; Alagmand et al., 2010; Sarhadi et al., 2012). Frequently, applications for flood hazard mapping found in the literature are based on low-resolution models, mainly using free digital elevation models, which are not vertically calibrated to a local geodetic reference system (Komi et al., 2017; Patel et al., 2017; Speckhann et al., 2017; Dhiman et al., 2019).

Thus, special attention is required for topographic data because a robust flood model needs high vertical accuracy linked to a geodetic reference system, since in many cases, flood-prone areas are found in marginal regions of low-slope floodplain and may embrace large areas, such as this case study (Uruguay River basin, Itaqui) (Gupta, 2009; Mistry, 2009). The modeling of a flood hazard map is linked to the local geodetic reference system, corroborated with the identification and monitoring of flood situations, and construction of realistic predictive scenarios (operational level) for risk management and adaptation (governmental agencies, such as the local civil defense) (Joshi et al., 2012; Gallien et al., 2018; Jongman, 2018). It is extremely important to do the calibration of the digital elevation model (DEM) using high-accuracy ground control point (GCP) data, aimed at improving the vertical accuracy, and applied to regional or local studies about floods (Araújo et al., 2018). Therefore, this study aims to conduct a robust mapping of flood hazard delimitation areas influenced by the Uruguay River, using the case study of Itaqui in the state of Rio Grande do Sul, southern Brazil, through a calibration of DEM from Shuttle Radar Topography Mission (SRTM) images, historical fluviometric level data collected at Itaqui station, and geoprocessing techniques.

## 2   Study area

The Uruguay River basin is one of the most important hydrographic basins of Brazil; it is located in the south of the country and extends throughout the neighboring countries of Argentina and Uruguay. Therefore, this basin marks the division between the Brazilian states of Rio Grande do Sul and Santa Catarina, and also between Brazil and Argentina. The Uruguay River basin occupies a total extension area of $385\,000\,\mathrm{km}^2$, where approximately $274\,300\,\mathrm{km}^2$ is located in Brazil, corresponding to 3 % of national territory. This hydrographic region has a prominent economic sector, mainly focused on the improvement of agricultural and industrial activities. The Uruguay River hydrographic region has great hydroelectric potential with a total production capacity of $40.5\,\mathrm{kW\,km}^2$, and when considering both Brazilian and Argentinian sides, one of the biggest energy per square kilometer relations in the world (ANA, 2015).

The Uruguay River basin area has a temperate climate, with a regular intra-annual rainfall distribution, and some highs during May to September, coinciding with the winter season in the Southern Hemisphere. The natural hydric availability of the Uruguay hydrographic basin is largely influenced by the significant spatial and temporal variables of a few climatic parameters, such as the pluviometric regime, which reflects directly on the economic activities developed in the region, which are largely in the agricultural sector (BID, 2008). Many areas were deforested due to the growth of agro-industrial activities in the region, which led to an environmental imbalance, from river siltation to water pollution with pesticides and their residues. Therefore, this region shows the original and remaining native vegetation cover of the Brazilian Atlantic forest biome and Araucaria moist forest biome. Geomorphology is dominated by rugged relief in the upper Uruguay River basin, followed by a flatter patch in the gaucho campaign region with shallow soil, which is the reason why the Uruguay River flows through a rocky substrate. This characteristic implies a flow regime that follows the rainfall trend: when periods of intense precipitation occur, they cause floods in riverside areas and, in the same way, when periods of drought occur, the flow regime is abruptly reduced, sometimes even to guarantee the hydric requirements of human and socioeconomic activities. In Brazil, the Uruguay hydrographic region is composed of 11 hydrographic sub-basins: (1) Apuae–Inhandava, (2) Passo Fundo, (3) Turvo–Santa Rosa–Santo Cristo, (4) Piratinim, (5) Ibicui, (6) Quarai, (7) Santa Maria, (8) Negro, (9) Ijui, (10) Varzea and (11) Butui–Icamaqua (ANA, 2015) CE2.

The study area comprises the urban area of Itaqui and is located on the banks of the Uruguay River on the western border of the state of Rio Grande do Sul, southern Brazil (Fig. 1). It corresponds to Ibicui sub-basin, the largest sub-basin of the Uruguay River, which has the Uruguay River and also the Cambai and Sanga das Olarias streams as its main water bodies. The study area has a territorial area of $34\,066.06\,\mathrm{km}^2$

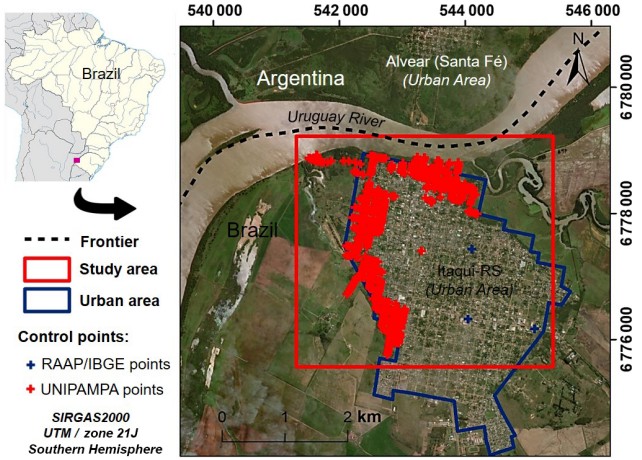

**Figure 1.** Location of the study area. Highlighted are the ground control points of high vertical accuracy from the High Accuracy Altimeter Network (in Portuguese, Rede Altimétrica de Alta Precisão – RAAP) of the Brazilian Institute of Geography and Statistics (IBGE) and from the Federal University of Pampa (UNIPAMPA).

and an estimated population of 39 049 inhabitants according to the Brazilian official 2016 Census by the National Institute of Geography and Statistics (IBGE, in Portuguese). This area plays an important role in regional but also national eco-
5 nomic scope due to rice cultivation, as well as Hereford and Braford bovine breeding, and commercial pig breeding for slaughter. These economic activities influence the region's environmental quality because they are directly related to the land use, occupation of the area, and increases in water con-
10 sumption (Bariani and Bariani, 2013).

Itaqui was settled on the edge of the Uruguay River. Currently, this riverside location is occupied by communities, some of very low economic class, that depend on the river for their livelihoods. These dwellers are often displaced to
15 higher regions to move away from the constant flood. Itaqui survives under the same conditions as most Brazilian small cities, with a lack of resources and of specialized professionals that can act in the projection and the orderly urban development. This fact is reflected in the lack of studies and
20 information that aid the aim of urban management to minimize negative impacts on society.

This undue occupation causes various problems that have a direct impact on society, including floods within the urban area. One of the reasons is the disorderly urban sprawl with-
25 out adequate infrastructure, along with the non-observance of the natural characteristics of the occupied environment that, as a consequence, means that it is impossible for areas near rivers and creeks to absorb floods (Silva et al., 2017). It is important to highlight that, in addition to the lack of financial
resources to be invested in the municipality, there the almost constant expense of the damages caused by the increase in river level. Most riverside families live in so-called "volantes

houses" (Fig. 2); these are wooden houses and can be transported from one place to another. But in some extreme events such as the one that occurred in 2017, some families lost all
35 of their belongings and in some cases even their homes, leaving it to the local government to help with the expenses of these communities (Fig. 3).

## 3 Previous studies of flooding in Itaqui

The flooding process of the Uruguay River in Itaqui is a
40 natural phenomenon that has afflicted the riverside population for decades. It is practically intrinsic to the city's history. However, even though it is a relevant problem to the local population, only after 2011 were papers published emphasizing the local hazards and risks of this natu-
45 ral phenomenon. Among the published papers, we emphasize Saueressig (2012), Saueressig and Robaina (2015), Silva et al. (2017), and Silva (2017). All of these are results of Masters dissertations in postgraduate programs in Brazil.

Saueressig (2012) emphasized a zoning of flood risk areas
of the urban marginal area of the Municipality of Itaqui. The author organized an inventory of floods that happened between 1980 and 2010, and focused on a socio-environmental discussion, developing models with a few consistent criteria in relation to altimetry.
Silva (2017) concentrated on the usage of geodetic methods to elaborate a DEM of integrated elevation with hydrologic data for monitoring affected areas lined by the Uruguay River. However, Silva (2017) focused his efforts on the marginal urban areas of Itaqui.
Thus, this work intends to fill an information gap, covering the entire urban area of the city, with the hazard in focus, priming a methodological application that uses high-accuracy altimetric data to model flood hazard maps. Furthermore, a statistical analysis using a temporal series of maxi-
mum annual level records of Uruguay River data was developed, which will provide innovative information about the return period of the flood phenomenon in study region.

## 4 Material and methods

The temporal series of annual maximum fluviometric records
of the Uruguay River, from 1942 to 2017 (76 years of data), linked to Brazilian Geodetic System (Sistema Geodésico Brasileiro – SGB) and using geometric leveling, was submitted for statistical analysis. Then, calibration of a digital elevation model (DEM), obtained free of charge from the Shuttle
Radar Topography Mission (SRTM), was conducted using ground control points (GCPs) of high vertical accuracy. The results of the temporal series statistical analysis served as the basis for the spatialization of floods on the calibrated DEM. Finally, a visual comparison between a modeled flood and a
digitally processed image from the flood area satellite image

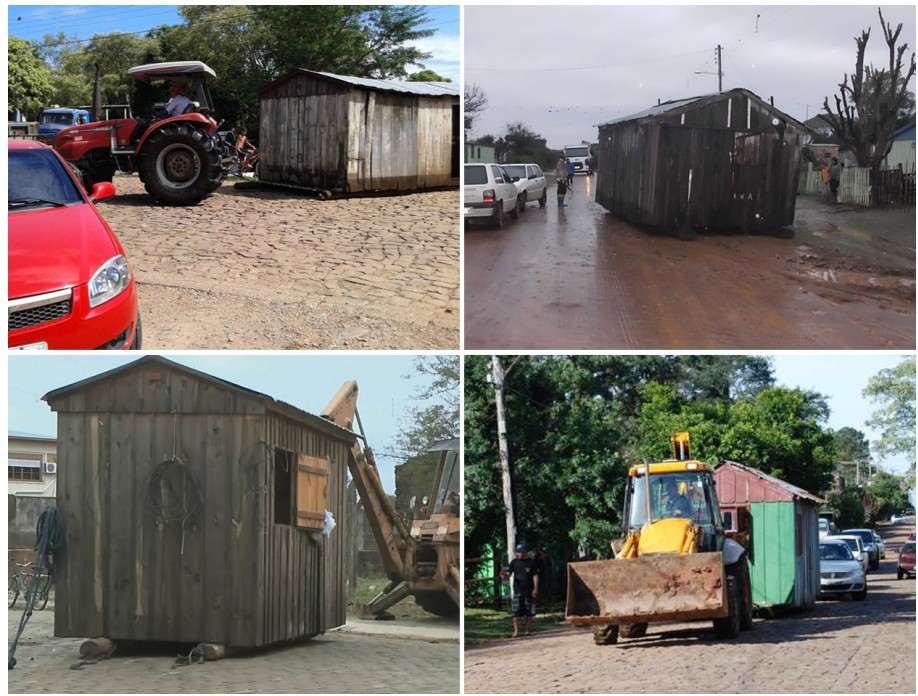

**Figure 2.** "Volantes houses" being transferred in periods of flooding in Itaqui.

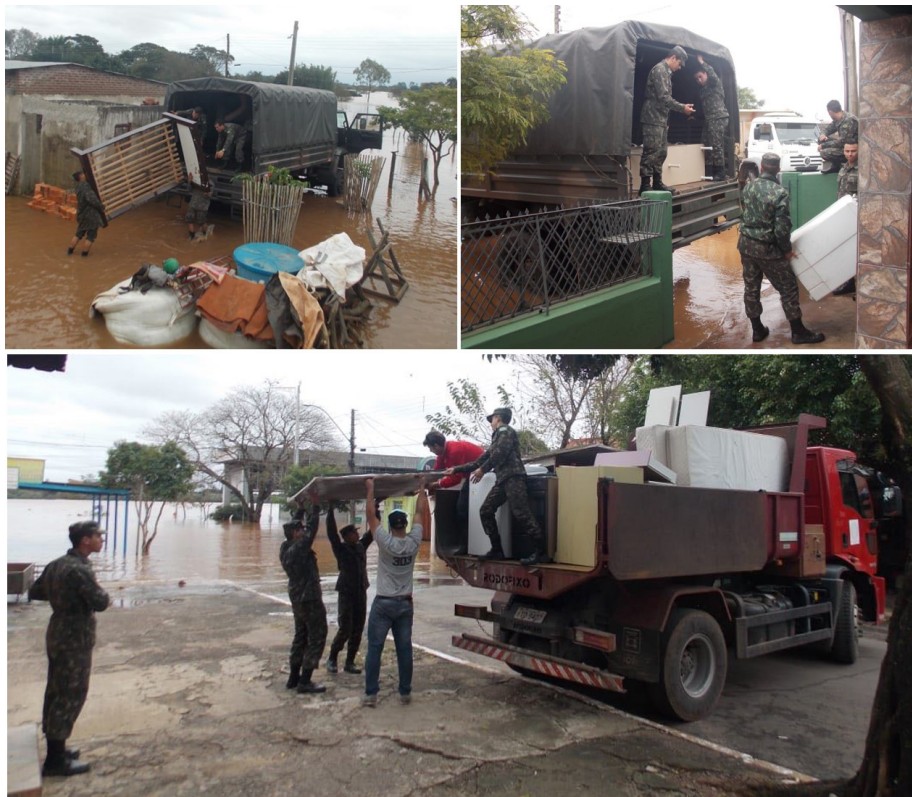

**Figure 3.** The Brazilian Army helping local residents during flooding events in Itaqui (adapted from: EB, 2017).

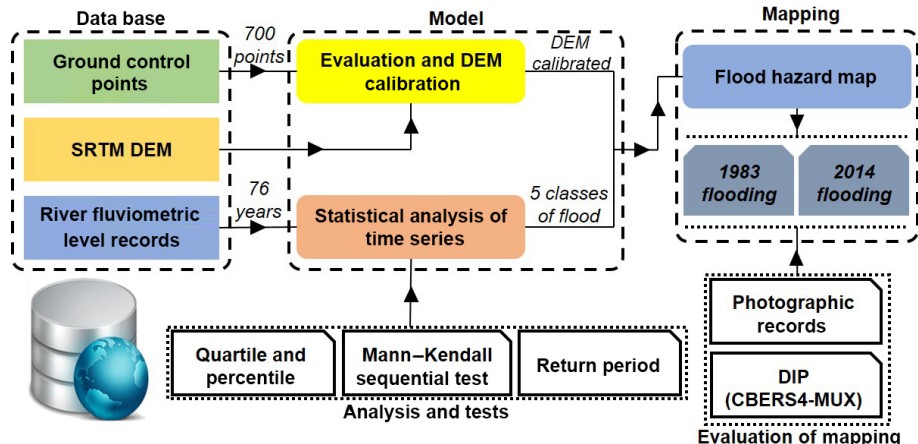

**Figure 4.** Flowchart of the proposed approach for the delimitation of flood hazard mapping. CE3

was performed, both having been registered concomitantly on the same day in study region.

In summary, the flood simulation model is based on the fill of the DEM, calibrated at river level orthometric heights, and 5 linked to a common geodetic reference system. The workflow of the proposed methodology is shown in Fig. 4.

### 4.1 Statistical analysis of Uruguay River fluviometric level records

Data from all annual maximums of fluviometric levels (m) 10 of the Uruguay River were assembled in the river level monitoring station with the codename "ITAQUI" no. 75900000 (latitude: $-29°07'39''$; longitude: $-56°33'45''$), for 76 years of historical information (between 1942 and 2017). All annual maximum level records were linked to the Brazilian 15 Geodetic System (Sistema Geodésico Brasileiro – SGB) with their respective orthometric heights. These data were obtained through the Monitoring and Alerts of Disasters System of Rio Grande do Sul (Sistema de Monitoramento e Alerta de Desastres – SMAD), using their website http://www.smad. 20 rs.gov.br/ ( TS4 ). The SMAD is a project implemented by the Environment and Sustainable Development Secretariat of Rio Grande do Sul, which is used by the civil defense and other competent organizations for monitoring and disaster warnings. Also, it is used for environmental management 25 and monitoring of natural resources.

In statistical analysis, first the variability of attributes and the characterization of the probability distribution were verified, based on the data descriptive analysis of the annual maximum orthometric heights of the Uruguay River. This analy-30 sis sought to obtain information about the central tendency, dispersion, and separatrix (quartile and percentile). Furthermore, the return period of maximum altimetric quotas for 2, 4, 10, 20, and 100 years were obtained. The return period ($T_{\mathrm{r}}$ TS5 ), also known as recurrence interval or recurrence time, 35 was employed as the time which a specific hydrological event

can be matched or exceeded in any given year (McCuen, 1998). In the present study, the return period ($T_{\mathrm{r}}$), in years, was defined by the following equation, where $p$ is the probability that a hydrological event be matched or exceeded (Tanguy et al., 2017), Eq. (1): 40

$$T_{\mathrm{r}} = \frac{1}{p}. \tag{1}$$

Then, the Mann–Kendall sequential test (Mann, 1945; Kendall, 1975) was applied to evaluate the temporal serial behavior of annual maximum orthometric height data of the Uruguay River. The Mann–Kendall test is a robust, sequen-45 tial, and non-parametric statistical method used to determine if a specific data series has a temporal tendency towards statistically significant changes. Among its advantages, it does not require normal distribution of data and is only slightly influenced by abrupt changes or non-homogenous 50 series (Zhang et al., 2009). In recent years, with growing concerns over environmental degradation and about the implications of greenhouse gases on the environment, researchers and practitioners have frequently applied the non-parametric Mann–Kendall test to detect trends in recorded hydrologic 55 time series such as water quality, streamflow, and precipitation time series (Yue and Wang, 2004). Although it has no influence on the flood hazard mapping, the Mann–Kendall test was applied to investigate if the elevation of the Uruguay River is showing any upward or downward trend. 60

### 4.2 Delimitation of flood hazards areas

#### 4.2.1 Digital elevation model (DEM) calibration

A digital elevation model (DEM) is a set of digital data describing elevation values of Earth ground surface (or any other surface) which contains additional information about 65 the character of this surface and interpolation algorithm, which is best for approximation (modeling) of the real topography (Szypuła, 2017). A DEM is a complete represen-

tation of a land surface, which means that heights are available at each point in the area of interest (Hengl and Evans, 2009). In this study, the SRTM DEM was taken as the topographic model. However, it was submitted to the calibration process for adjustment to the local reference geodetic system. The DEM calibration is a mandatory pre-processing adoption that provides improvement of both DEM vertical accuracy and linking to the geodetic system (Araújo et al., 2018). This study adopted the SRTM calibration method suggested by Araújo et al. (2018). These authors used control points of high vertical accuracy linked to the Brazilian Geodetic System (in Portuguese Sistema Geodésico Brasileiro – SGB) for calibrating the DEMs.

The GCPs were considered as the orthometric altimetry points acquired from high-accuracy geodesy. A grid with 700 GCPs was constructed using data from two databases (Fig. 1): (i) 3 points from the High Accuracy Altimeter Network (Rede Altimétrica de Alta Precisão – RAAP) of the Brazilian Institute of Geography and Statistics (IBGE); and (ii) 697 points with orthometric heights linked to SGB through relative GNSS leveling, called UNIPAMPA points.

The RAAP–IBGE was created using the high-accuracy geometric leveling technique, allowing the determination of geodetic stations with an altitude value that was referenced with the mean sea level (MSL) in Imbituba-SC. These stations are known as level references (Referencias de Niveis – RRNN) and were set up throughout Brazil along highways and railways at around 3 km intervals, in the first survey, and nowadays at around 2 km intervals. Currently, the Brazilian network has approximately 68 000 RRNN available at the Geodetic Database (Banco de Dados Geodesicos – BDG), which can be accessed at the Brazilian Open Data website (Portal Brasileiro de Dados Abertos – PBDA), through the following link: http://dados.gov.br/dataset/cged_bdg_rn ( TS6 ). For this study, three points from RAAP that were inside the study area and in a good conservation state were used.

The database with 697 control points from the Federal University of Pampa (UNIPAMPA) was acquired with GNSS receivers on the field through a high-accuracy post-processed kinematic (PPK) mode and linked to SGB (Silva et al., 2017). These high-accuracy control points had a mean altimetric error of $0.006 \pm 0.0007$ m.

All 700 control points were standardized to the horizontal datum SIRGAS2000, the Universal Transverse Mercator coordinate system (UTM, Zone: 21J/Southern Hemisphere), and the MAPGEO2010 geodetic model. Lastly, the orthometric height values were exported in a shapefile format with the ArcMap 10.1 software (ESRI, 2011).

The DEM used in this study was obtained through a SRTM image with a spatial resolution of 1 arcsec (approximately 30 m at the Equator), 16 bit radiometric resolution, a geographic coordinate system, WGS1984 horizontal datum, and a EGM96 vertical datum (Earth Gravitational Model, 1996). These data were acquired at the Earth Data servers (NASA),

using the interface developed by Derek Watkins (http://dwtkns.com/srtm30m/, TS7 ). The advantage of this interface is the availability of SRTM scenes with excellent void correction processes, which are empty spaces where no elevation value can be determined. These voids cause significant problems for using the DEM from SRTM images, especially on hydrological modeling applications that require continuous flow surfaces. The "S30W057" scene was downloaded and the orthometric height values of pixels were extracted from the control point grid and exported in table form for statistical analysis purposes.

To evaluate and calibrate the DEM of the study area, a matrix was constructed with the orthometric height values of control points and the SRTM image. This dataset was subjected to linear regression analysis, with ground control point values as a dependent variable and SRTM data as an independent variable, which is a common procedure found in the literature for DEM calibration (e.g., Gorokhovich and Voustianiouk, 2006; Forkuor and Maathuis, 2012). Then, the DEM obtained from the SRTM image was calibrated using the model proposed by the linear regression. Subsequently, both DEMs (the original and calibrated) were subjected to descriptive statistical analysis and comparison between the mean of orthometric height variation ($\Delta H$) and the root mean square error (RMSE) value, as proposed by Araújo et al. (2018). In all statistical analyses, a significance value of 5 % (Zar, 2010) was adopted and they were performed on R software v.3.4.1 (R Development Core Team, 2017).

### 4.2.2   Determining classes of flood hazard map

To determine the classes of flood hazard mapping, a descriptive analysis of the orthometric height temporal series (annual maximum fluvial levels records) of the Uruguay River was performed (minimum, maximum, quartile, and percentile). The determination of the classes was closely linked to the probability of occurrence of the annual maximum fluvial height of the Uruguay River. At this stage, five classes of flood hazard were determined, as described in Table 1.

Through map algebra, using the previously calibrated DEM, all classes were mapped in a geographic information system (GIS) environment using ArcMap 10.1 software (ESRI, 2011).

In a few areas there was topographic discontinuity. When these altimetric class discontinuities occurred, the sector in focus was considered to be in the upper elevation altimetric class.

### 4.2.3   Evaluation of the mapping of flood hazard areas

At this stage, the flood area for the 12 June 2017 was estimated using the flood model established in the previous stage. This date was picked because it was the annual maximum level day of the Uruguay River for the year 2017 in the

**Table 1.** Classes of flood hazard for mapping.

| Classes | Altimetric river quota used as indicator |
| --- | --- |
| Extremely high flood hazard | < Median |
| High flood hazard | ≥ Median and third quartile |
| Moderate flood hazard | third quartile and < 95 % |
| Low flood hazard | ≥ 95 % and < maximum quota |
| Non-floodable | > Maximum quota |

**Table 2.** Results of Mann–Kendall test.

| Summary | |
| --- | --- |
| Kendall's tau statistic | 0.167 |
| Two-sided $p$ value | **0.03351** TS9 |
| Kendall score ($S$) | 475 |
| Denominator ($D$) | 2848.5 |
| Variance of Kendall score | 49 713.67 |

**Table 3.** Probability of the fluviometric level being matched or exceeded. $T_r$: flood return period.

| Probability ($p$) | $T_r$ (years) | Altimetric river quota (m) |
| --- | --- | --- |
| 0.5 | 2 | 51.66 |
| 0.25 | 4 | 53.45 |
| 0.1 | 10 | 54.58 |
| 0.05 | 20 | 55.92 |
| 0.01 | 100 | 57.24 |

Itaqui city area. In addition, it coincided with the CBERS-4 (China–Brazil Earth Resources Satellite) satellite multispectral imagery for the region. Among the coupled instruments in CBERS-4, the Multispectral Regular (MUX) sensor stands out, covering four spectral bands between 450 and 890 nm, with a scanning range of 120 km, and nominal spatial resolution of 20 m at nadir (Boggione et al., 2014). These data were acquired freely from the National Institute of Space Research (Instituto Nacional de Pesquisas Espaciais – INPE) by accessing their website (http://www.inpe.br/, TS8). The multispectral bands were subjected to a radiometric and atmospheric correction (Jensen, 2009). A hybrid color composite RGBI-765NDWI, based on a red–green–blue intensity color system, was generated in order to identify the flood area, using ERMAPPER v7.1 software (ERDAS, 2008). The normalized difference water index (NDWI) was elaborated using the following algorithm proposed by McFeeters (1996), Eq. (2):

$$\text{NDWI} = \frac{\rho_{\text{green}} - \rho_{\text{NIR}}}{\rho_{\text{green}} + \rho_{\text{NIR}}}, \tag{2}$$

where $\rho_{\text{green}}$ corresponds to the reflectance of the green band (CBERS-4 band 6; 520–590 m); and $\rho_{\text{NIR}}$ corresponds to the reflectance of the infrared band (CBERS-4 band 8; 770–890 m) of sensor CBERS-4 MUX.

Finally, a visual comparison of the modeled flood area vs. identified flood area on satellite image was performed.

## 5   Results and discussion

In this work the use of geoprocessing techniques was extremely important to reach the results. We use the following techniques:

- Geographic information system (GIS) was the technique most used throughout the work. All data were used to implement a robust GIS.

- Digital cartography was used during the elaboration of the maps.

- Digital image processing (DIP) CE4 was the technique applied to improve the visualization of the historical flooding from the CBERS-4 MUX satellite images.

- Precision geodesy was used during the calculation of the points of land controls and the linkage of the river level to the Brazilian Geodetic System.

- Geostatistical techniques were used during the evaluation and calibration of the digital elevation model.

The results achieved in this study are systematically presented and discussed below.

### 5.1   Analysis of the maximum annual fluviometric level of the Uruguay River

The annual maximum level records of the Uruguay River for the last 76 years have an amplitude of 9.78 m, with a maximum of 57.23 m and a minimum of 47.76 m. Furthermore, the values of mean and median were 51.97 and 51.56 m, respectively (Fig. 5). In the Mann–Kendall test, the temporal series showed no tendency (two-sided $p$ value = 0.03351), as shown in Table 2. However, it is essential to mention that, during the last 3 years, two occurrences of a flood altimetric quota for a return period of 20 years were observed (2014 and 2017). The return period values for 2, 4, 10, 20, and 100 years are presented on Table 3.

It is important to emphasize that the Uruguay River has large flow variation, from the occurrence of great floods, which affect riverside populations, to a lack of water for human consumption and other necessities. This seasonal variation of hydric availability is caused, in general, by the low permeability of the soils (shallow and rocky), which accumulates only a little water and does not support an adequate base

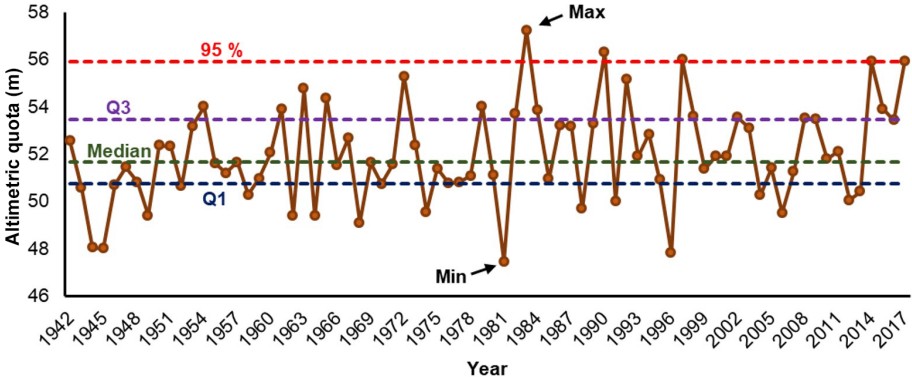

**Figure 5.** Annual fluviometric maximum level records of the Uruguay River in Itaqui.

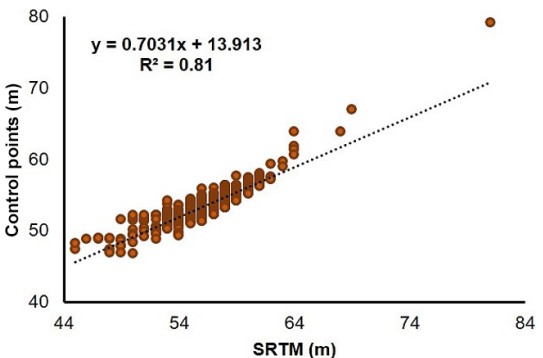

**Figure 6.** Linear regression analysis of SRTM image vs. ground control points (GCPs).

**Table 4.** Indicators of DEM errors in relation to GCPs.

| DEM | Mean of $\Delta H$ (m) | RMSE (m) |
|---|---|---|
| SRTM$_{Original}$ | −2.84 | 3.14 |
| SRTM$_{Calibrated}$ | 0.00 | 1.00 |

**Table 5.** Classes of flood hazard mapping in Itaqui, southern Brazil.

| Class | Altimetric river quota used as indicator |
|---|---|
| Extremely high flood hazard | < 51.66 |
| High flood hazard | 51.66|–53.45 |
| Moderate flood hazard | 53.45|–55.92 |
| Low flood hazard | 55.92|–57.24 |
| Non-floodable | > 57.24 |

hydric flow sufficient for dry periods. Furthermore, physical characteristics of the Uruguay River basin, such as the strong slope of the drained terrain, river channel morphology, and the soil, in conjunction with low water retention and deforestation of the main river and tributary banks for the extraction of wood and/or agricultural use, cause rapid surface drainage after intense rains, flooding, erosion, and silting of river beds (BID, 2008).

## 5.2 Delimitation of flood hazard areas

Using the 700 GCPs, it was possible to evaluate and calibrate the DEM from the SRTM image. The linear regression analysis showed strong correlation between the ground control points and the SRTM altimetry data, generating a highly robust model with $R^2 = 0.81$ and $p < 0.001$ (Figs. 6 and 7). The SRTM calibration and correction model based on control points was determined as $y = 0.7031x + 13.913$.

Comparing the error evaluators, it was already expected that the calibrated SRTM mean of the orthometric height variation ($\Delta H$) would be zero (Table 4). According to Araújo et al. (2018) this happens due to the DEM plan being perfectly adjusted to the GCP network plan after calibration, linking the MDE calibration to the Brazilian Geodetic System (SGB). In relation to the RMSE of the original SRTM (non-calibrated SRTM), a RMSE of 3.14 m was found from comparison to the GCPs. This accuracy is approximately 2 times greater than what was reported in the original SRTM specifications (6.20 m for South America). Araújo et al. (2018) found a similar value (3.10 m) when evaluating the DEM from SRTM at regional-scale analysis for the Piranhas–Açu River basin, in northeastern Brazil. When calibrating the SRTM, an extreme improvement of the RMSE was verified, where the adjusted value of 1.00 m was found. In this way, a 68.15 % decrease of RMSE value for the calibrated SRTM compared to the non-calibrated SRTM was found (Table 4).

Assuming that the calibrated SRTM is linked to the SGB, it was possible to apply the five flood hazard classes to the altimetric model, as shown in Table 5. After the spatialization of the classes it was possible to observe the flood behavior in the study area (Fig. 8).

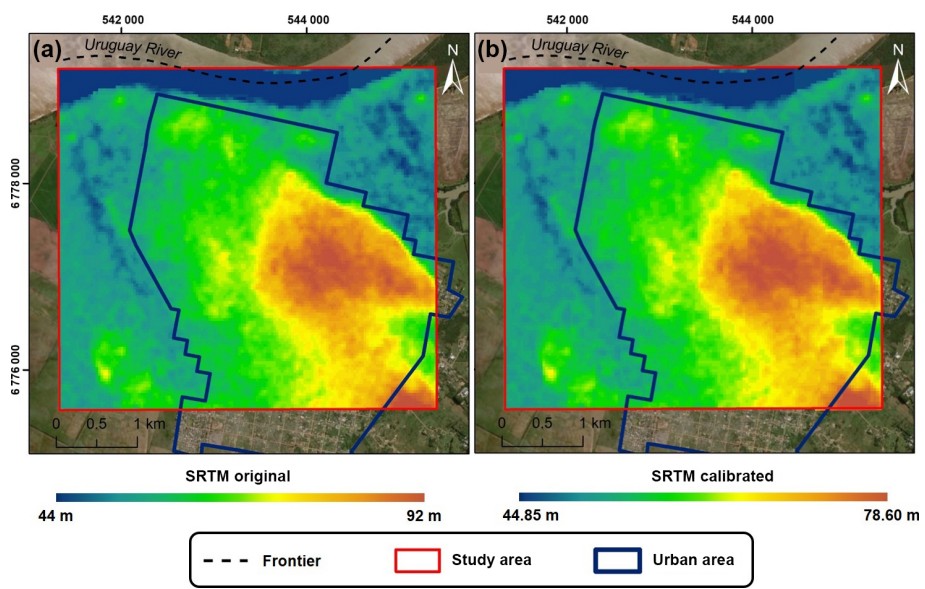

**Figure 7.** Digital elevation model (DEM): **(a)** SRTM original; **(b)** SRTM calibrated.

Mapping results of the flood hazard classes helped to identify areas with greatest potential for damage to the population in the study area. In general, there was a river gradient upstream from the urban city center, mainly around the north and west neighborhood outskirts of the urban area, with greater emphasis on the Cerrinho Dois Umbus, Centro, Ponte Seca, Varzea, Enio Sayo, and Capelinha neighborhoods CE5 (Fig. 8).

The methodological approach and resulting flood hazard map highlighted flood-prone areas throughout the municipality of Itaqui that have the potential of being exposed to flooding events, inflicting suffering on the population and substantial material damage. Figure 9 illustrated the boundaries of the Uruguay River flooded area for a 100-year return period ($T_r = 100$ years), obtained from the proposed model, also indicating flood-prone areas. This return period shows the same flood pattern that occurred in July 1983 when Itaqui suffered its worst historical flood (altimetric level $= 57.24$ m). According to Saueressig and Robaina (2015), the water level reached the Marechal Deodoro Square, the courtyard of the Itaqui prison (Fig. 9), and most of the neighborhoods bordered by the Olaria stream.

Geoprocessing techniques were successfully used to visualize the extent of flooding and also to produce a significant improvement in forecasting flood hazard maps, but can be also employed to establish a decision-based support system through a partnership between scientists, territorial planners, and policy makers (Wiles and Levine, 2002; Sole et al., 2007; Korah and López, 2015; Demir and Kisi, 2016; Sahoo and Sreeja, 2017). The hybrid color composite RGBI-765NDWI also highlighted suspended sediment in the water column of the Uruguay River and its tributaries. The result is a scene with intrinsic differences in water color with regard to water

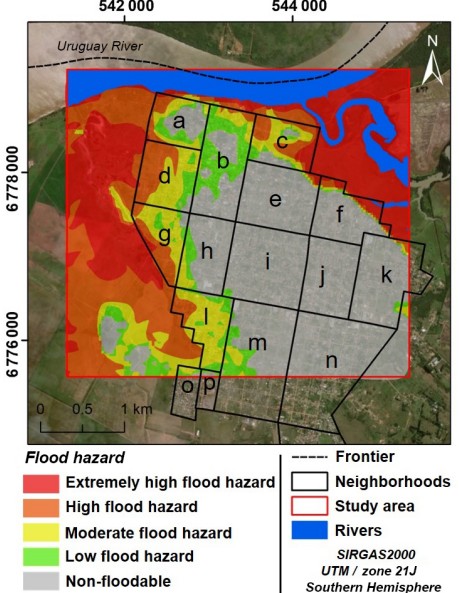

**Figure 8.** Flood hazard map of Itaqui, Rio Grande do Sul, southern Brazil. Neighborhoods: (a) Cerrinho Dois Umbus; (b) Centro; (c) Ponte Seca; (d) Varzea; (e) Cidade Alta; (f) Cohab; (g) Enio Sayago; (h) Estacao; (i) Capelinha; (j) Dr. Ayub; (k) Jose da Luz; (l) Vinte e Quatro de Maio; (m) Chacara; (n) Cafifas; (o) Vila Nova; and (p) Uniao. CE6 CE7

quality and it therefore benefits from precise demarcation between the inundated and relatively dry areas, i.e., "flooded" and "non-floodable" classes, respectively. Comparison between the flood simulation model for the 12 June 2017 annual maximum level record and the CBERS-4 MUX satellite image for the same date showed that similar spatial behav-

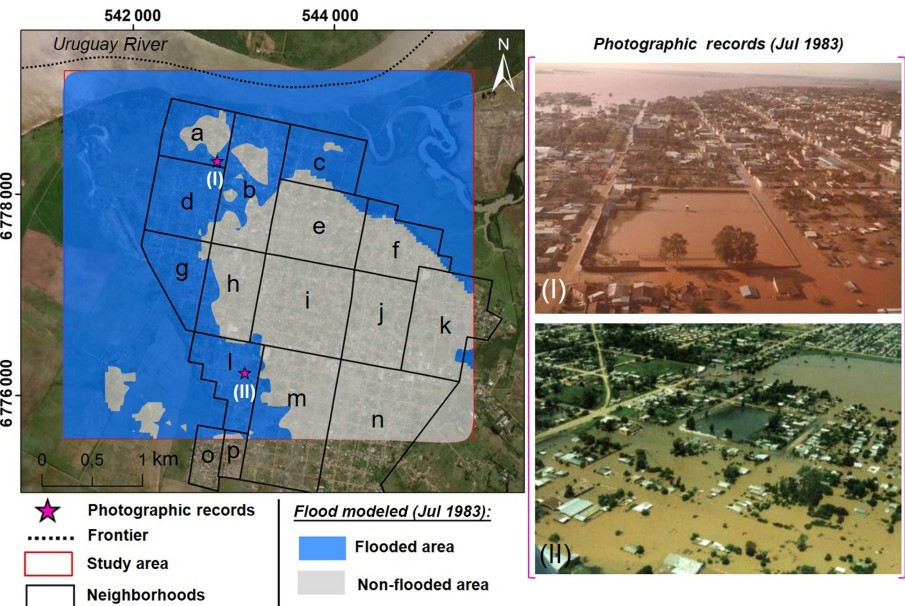

**Figure 9.** The worst historical flooding of the Uruguay River basin (July 1983) has been used here as a base for a simulated flood event in Itaqui. Neighborhoods: (a) Cerrinho Dois Umbus; (b) Centro; (c) Ponte Seca; (d) Várzea; (e) Cidade Alta; (f) Cohab; (g) Enio Sayago; (h) Estacao; (i) Capelinha; (j) Dr. Ayub; (k) Jose da Luz; (l) Vinte e Quatro de Maio; (m) Chacara; (n) Cafifas; (o) Vila Nova; and (p) Uniao. Photographic records: **(I)** the courtyard of the prison of Itaqui (Flores and Flores, 1983; Saueressig e Robaina, 2015); **(II)** the neighborhood of Vinte e Quatro de Maio (Boeira, 1983; Saueressig, 2012). CE8 CE9

ior was verified in both flooded areas (Fig. 10). A similitude is prominent between the result of the flood modeling and the real flooded area based only on visual analysis (Fig. 10a and b, respectively).

Mukolwe (2016) emphasized that supporting strategies for flood hazard mitigation, or even a sufficient understanding of the application of their spatial distribution, is a relevant enterprise for assisting government management and risk planning. In practice, the methodological strategy and the resulting models presented in this study can be used for the operational real-time phase of flood monitoring during seasonal flood events in the Uruguay River basin. Such a methodological approach based on control by the local civil defense will allow residents to be evacuated from the areas that will be affected. Thus, the communities who reside in "volantes houses" can be moved to places further from the flood hazard area, and those who reside in masonry houses can be removed and sheltered, in addition to reducing damage costs by minimizing losses of belongings.

## 6    Conclusions

A large part of the city of Itaqui is in floodplain area of the Uruguay River, as are many of the riverside cities of the study area (Saueressig, 2012). The floods in these regions are intimately linked to the rise of the river level. Therefore, in the case under study, the river level altimetry is the main driver of the flood hazard.

Flood hazard maps are crucial for planning and intervention in flood-prone areas. The results of the Itaqui case study can mainly be applied to characterizing hazards and supporting the implementation of flood risk management plans and flood risk maps for river basins and coastal areas, thus improving the overall availability of such risk management tools. An equivalent methodological approach could be helpful, for instance, in preparing urban charts to identify areas more suitable to occupation in the municipality. In cases where the areas are already occupied, like in some locations of the Itaqui city center, the result could be useful for defining and implementing the necessary measures that address potentially damaging events.

In Brazil, only a few studies involve mapping to assess potential flood damage. Spatial details of the hazard indicators are a valuable tool for flood risk management since the map provides a more direct and fast assessment than other methods. The methodology and materials applied to Itaqui proved effective in identifying flood hazard areas using free DEM from SRTM images that were calibrated based on a post-processed high-precision GNSS database, historic river level gauge data, and geoprocessing techniques, as basic information.

It is worth mentioning that the methodological approach applied to the Itaqui is adequate to be replicated to other municipalities, in particular those of riparian and coastal communities. The flood hazard mapping methodology could be particularly useful for regions with a good historic time series

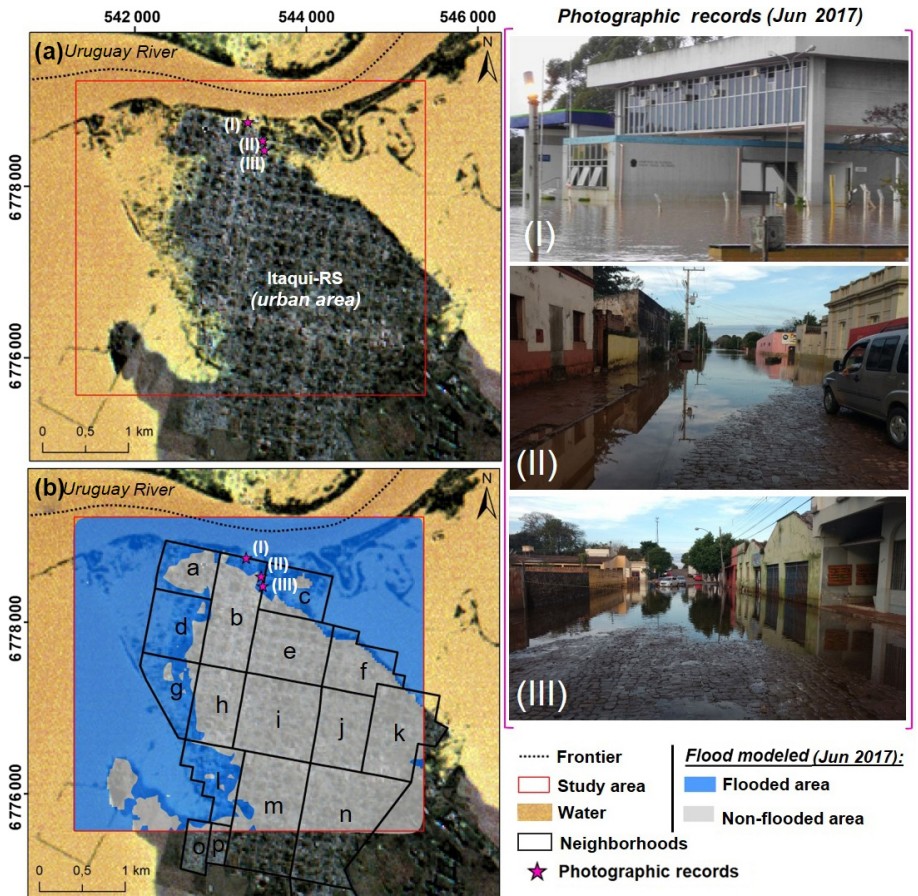

**Figure 10.** The 2017 historical flooding of the Uruguay River basin: (a) CBERS-4 MUX satellite image showing the extent of the floods in the Itaqui city area on 12 June 2017 (hybrid color composition: RGBI-765NDWI); and (b) simulated flood event in Itaqui for the same satellite imaging date. Neighborhoods: (a) Cerrinho Dois Umbus; (b) Centro; (c) Ponte Seca; (d) Varzea; (e) Cidade Alta; (f) Cohab; (g) Enio Sayago; (h) Estacao; (i) Capelinha; (j) Dr. Ayub; (k) Jose da Luz; (l) Vinte e Quatro de Maio; (m) Chacara; (n) Cafifas; (o) Vila Nova; and (p) Uniao. Photographic records: **(I)** the Federal Revenue Customs Building; **(II)** Osvaldo Aranha street (Silva, 2017); **(III)** Borges do Canto street (Silva, 2017). CE10 CE11

of fluviometric level gauge data. In general, the most limiting factors to be considered for adopting these methodological approaches are currently the availability of high-accuracy altimetry points, the GCPs, for calibrating the free DEM from the SRTM scene in regions impacted by a series of flood events. Such a procedure involves a large amount of time, both for the collection of local GCP data and also for postprocessing data treatment. However, depending on the geographic location, IBGE level references (RRNN) can be used as GCPs.

This flood hazard mapping in digital format can be used as a database to assist governmental stakeholders, e.g., civil defense, in partnership with other sectors of civil society for implementation of flood risk management plans that are more adaptable to local restrictive environmental constraints. Thus, composing these plans can be more flexible, easily modifiable, and updatable in accordance with the dis-

tinguishing physiographic and socioeconomic setting of the river basin.

In this distinct case study, the scenario modeling for the 1983 and 2017 flooding events were compared with a real flood of 12 June 2017 and high visual similarity was established between them. The study area generally covers residential areas. Considering the results of this case study, it is crucial to advise the Itaqui policy makers and politicians that a planned public policy should be implemented to relocate part of the urban population that occupy areas of high flood hazard. However, the removal process of inhabitants to other areas without hazard, or even less risk, is complex and requires a multidisciplinary strategy. It is already known that the population will suffer historical, social, and cultural impacts that will make the planning more difficult to be entirely applied. Therefore, it is necessary that people have decent housing, with adequate infrastructure and basic sanitation, reducing their exposure and vulnerability to risk adversity.

Furthermore, it is necessary for communities to understand the importance of these changes, not only for the population's health but also for the Itaqui economy and sustainability.

This paper also demonstrates that well-applied technical measures based on geotechnologies, such as remote sensing, GIS, and high-accuracy geodesy, give results in return that can be very effective in urban and rural management with low-cost investments, highlighting the unique features of a given locality, especially floodplains and flat low-lying areas. This methodological approach is very effective for mitigating flooding damage in coastal and riparian areas. It can be valuable in reducing strategic monitoring costs and the operational expenses of providing assistance to population affected by severe flooding events and their consequences.

*Data availability.*   . TS10

*Author contributions.*   . TS11

*Competing interests.*   The authors declare that they have no conflict of interest. TS12

*Special issue statement.* This article is part of the special issue "Flood risk assessment and management". It is a result of the EGU General Assembly 2018, Vienna, Austria, 8–13 April 2018. TS13

*Acknowledgements.* The authors express special thanks to the Secretary of Environment and Sustainable Development of Rio Grande do Sul for providing river level gauge data. We also thank the reviewers of the *Natural Hazards and Earth System Sciences (NHESS)* journal for their many insightful comments.

Edited by: Dhruvesh Patel
Reviewed by: Caterina Samela and one anonymous referee

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

Please note the remarks at the end of the manuscript.

**Remarks from the language copy-editor**

CE1     Please confirm change.
CE2     Please thoroughly check all Portuguese names for spelling.
CE3     Please note that this figure has undergone image reprocessing.
CE4     Please confirm change in acronym.
CE5     Please check all spellings for consistency and accuracy.
CE6     Please check all spellings for consistency and accuracy.
CE7     Please note that this figure has undergone image reprocessing.
CE8     Please thoroughly check all Portuguese names for spelling.
CE9     Please note that this figure has undergone image reprocessing.
CE10    Please thoroughly check all Portuguese names for spelling.
CE11    Please note that this figure has undergone image reprocessing.

**Remarks from the typesetter**

TS1     The composition of Figs. 1 and 4–10 has been adjusted to our standards.
TS2     Copernicus Publications collects the DOIs of data sets, videos, samples, model code, and other supplementary/underlying material or resources as well as additional outputs. These assets should be added to the reference list (author(s), title, DOI, and year) and properly cited in the article. If no DOI can be registered, assets can be linked through persistent URLs. This is not seen as best practice and the persistence of the URL must be secured.
TS3     The reference of Demir (2015) is missing in the reference list. Please check.
TS4     Please provide last access date.
TS5     "Tr" changed to "$T_r$" throughout the paper. Please check.
TS6     Please provide last access date.
TS7     Please provide last access date.
TS8     Please provide last access date.
TS9     Please give a description for this bold value.
TS10    Please provide a statement on how your underlying research data can be accessed. If the data are not publicly accessible, a detailed explanation of why this is the case is required. The best way to provide access to data is by depositing them (as well as related metadata) in reliable public data repositories, assigning digital object identifiers (DOIs), and properly citing data sets as individual contributions. Please indicate if different data sets are deposited in different repositories or if data from a third party were used. If no DOI is available, assets can be linked through persistent URLs to the data set itself (not to the repositories' home page). This is not seen as best practice and the persistence of the URL must be secured.
TS11    Copernicus Publications strongly recommends including the section "Author contributions".
TS12    Declaration of all potential conflicts of interest is required by us as this is an integral aspect of a transparent record of scientific work. If there are possible conflicts of interest, please state what competing interests are relevant to your work.
TS13    Please confirm.
TS14    Please provide last access date.
TS15    Please provide last access date.
TS16    Please provide last access date.
TS17    Please provide page range or article number.
TS18    Please provide place of publication or URL and last access date.
TS19    Please provide place of publication or URL and last access date.
TS20    Please provide place of publication.
TS21    Please provide place of publication.
TS22    Please provide last access date.
TS23    Please provide last access date.
TS24    Please provide place of publication.
TS25    Please provide last access date.
TS26    Please provide place of publication.