# Peer review of "Delimitation of Flood Areas Based on Calibrated DEM and Geoprocessing: Case Study on Uruguay River, Itaqui City, Southern Brazil"

_Natural Hazards and Earth System Sciences, 2018_

## Editor Comment (EC1) · D. Patel (Editor) · 20 Oct 2018

Comments to the Authors: [Paulo Victor. N. Araújo, Venerando E. Amaro, Robert M. Silva, Alexandre B. Lopes] We would like to thanks the authors for submitting their research finding in NHESS. Paper is as per the scope of the journal, however at preliminary review, Editorial panel has observed significant corrections, and some of them are cited here for author's consideration to improve the quality of work. Minor corrections Pg.2, Line 26, use the superscript function to write the unit i.e. km2 instead of km2. Pg. 3, Line 16, Official instead of oficial Use a common scale bar and Grid frame

[Figure]
Interactive
comment

to fig 1, fig 6, fig 7, fig . 8 (a), (b) Cite all the reference as per NHESS reference citation style Major comments 1) Authors has used flood simulation model, however the description of model to prepare the flood hazard map is missing. 2) Authors have utilized SRTM DEM 1 arc data set, Although DEM of a study area has not been provided in form of figure or map. It is important to include. 3) How the SRTM DEM is used to prepare the flood hazard maps? It is important to describe. 4) What is Geoprocessing? Which geoprocessing technique has been used by authors to prepare a flood hazard maps? Explain. 5) Any rational of selections of flood hazard classes in Table 1 and Fig 6, if so, please explain. 6) Table 2, Kendall's tau statistic 0.167, what its correlation with flood hazard mapping? , it is important to explain significance of Mann Kendall test in flood mapping. It is important to correct and resubmit the revise version.

---

## Referee Comment (RC1) · Ph.D. Samela (Referee) · 1 Nov 2018

Summary:

The authors propose a procedure for the identification of flood-prone areas using a Digital Elevation Model (DEM) calibrated with 700 Ground Control Points (GCPs), historical river level data (76 years of data), and geoprocessing techniques. The study area is a portion of the Uruguay River basin close to the city of Itaqui, Southern Brazil.

General Comment:

[Figure]

The topic is certainly of interest to the readership of this journal and the scientific community. However, I find several concerns that deserve to be addressed and I would like to encourage the authors to answer to the following questions or suggestions in order to promote the performed research before a possible publication in this journal. 1) The introduction is focused almost exclusively on the importance and the role of flood hazard maps. The background of the research has not been delineated. Traditional procedures for flood hazard studies, or alternative methods, and the problems/limits related to both of them are not mentioned. Therefore, it is not clear what is the gap or issue that the proposed research aim to address? In few words, the aims are not clearly defined.Also, data, methods, models, performance measures should be illustrated in more detail. 2) The extent of the study area is not clear. Section "Study area" reports that the full Uruguay basin has a total area of 385,000 km2. Then, it is also reported that the study area corresponds to Ibicui sub-basin, the largest Uruguay river sub-basin, and that this study area has a territorial area of 3,406,606 km2. How can this sub-basin have a drainage area larger than the full hydrographic basin? 3) A SRTM-DEM has been calibrated using ground control points (GCPs) of high vertical accuracy. Can you provide a quantitative indication about GCPs vertical accuracy? 4) In carrying out the visual comparison in Section 4, please, explain more clearly what do you mean with "simulated flood altimetric quota". More details as regards the simulation need to be provided. 5) Section "4.2.1 Digital Elevation Model (DEM) calibration" specifies that in performing the linear regression, GCPs values have been used as independent variable and SRTM data as dependent variable. The independent variable is usually a measurement you are not manipulating in your experiment, and conventionally it is on the x axis. Instead Figure 5 puts SRTM values on the x-axis. Can you clarify Figure 5, the linear function y=0.7031x+13.913 you derived, and how did you use it? 6) As far as I understand, the function obtained in Linear Regression has been used to predict the dependent variable values (the DEM values) as a function of the GCPs. Then the original SRTM DEM and the DEM adjusted with GCPs have been compared and RMSE has been evaluated. More interesting, in my opinion, would be to

make a statistical comparison between the "new" adjusted DEM values against GCPs different from the ones used for the calibration, in order to validate the improvement in accuracy produced by this procedure. 7) As regards the comparison showed in Figure 8 between the results of the proposed approach and CBERS-4/MUX satellite image for 12 June 2017, I suggest complementing this visual comparison with some statistics and performance measures. I believe this validation will improve the manuscript and the reliability of the proposed method.

Minor comments: I am not a native speaker, but in my opinion the paper needs thorough reading and correction of English language and technical language. Below a few examples that I found while reading the manuscript: 1) Line 2, Abstract: replace "historic" with "historical". 2) Line 15, Abstract: instead of "fluviometric temporal series records" I suggest "temporal series of streamflow records". 3) Line 15-16, Abstract: Check subject-verb agreement in "The annual maximum...were linked to..." 4) Line 16, Abstract: "submitted the statistical analysis". Unclear. 5) Line 18-19, Abstract: "Using the temporal series statistical analysis results, was assessed the spatialisation of flood hazard classes on the calibrated DEM and validated". Please, rephrase and move the verb "was assessed" after its subject "the spatialisation of flood hazard classes on the calibrated DEM". 6) Line 23, Abstract: instead of "Were determinate 5 classes of flood hazards", move the verb at the end of the sentence and correct it in "were determined". 7) Line 28, Introduction: check subject-verb agreement: "causes" instead of "cause". 8) Page 2, lines 3-7, Introduction: "These geohazards can be prevented and reduced by providing reliable information to the public about the flood hazard through flood inundation maps (Alaghmand et al., 2010; Demir, 2015). Information about the flood's extension is extremely important to evaluate the hazard of flood-prone areas and to help the rescue operations during these events (Cook and Merwade, 2009). Flood hazard mapping is one of the tools used to help communities avoid or mitigate such losses and damages (Arrighi et al., 2013; Savage et al., 2014; Speckhann et al., 2017). Flood hazards maps need therefore to be created as they provide a basis for the development of flood risk management plans". 9) Please, rephrase and avoid

repetition of the same concept. 10) Page 2, lines 15-16, Introduction: use "high vertical accuracy" in "topographic data... must possess vertical highly accuracy altimetric". 11) Page 3, line 16, Study area: "official" instead of "oficial". 12) Page 3, line 15, Study area: correct "rive" with "river". 13) Page 4, Line 4, Section "3 Previous studies in Itaqui city on flooding": Check subject-verb agreement and grammar in "The flooding process of Uruguay River in Itaqui city are a natural phenomenon that afflicts the riverside population for decades". 14) Page 4, Line 6, Section "3 Previous studies in Itaqui city on flooding": "risks" instead of "riscks". 15) Page 4, Line 15: "fulfill" instead of "fulfil". 16) Page 4, Lines 15,16: "priming in the use on the high elevation accuracy of altimetric and fluviometric data to the modelling of flood geohazard mapping". Revise English. 17) Page 4, Lines 20-22: "was submitted the statistical analyses". Unclear. 18) Page 5, line 26: "Was considered as GCPs only the orthometric altimetry points acquired from high accuracy Geodesy which data were based in Global Navigation Satellite System (GNSS)". Revise English and subject-verb agreement. 19) Page 6, lines 28-29: "temporal series descriptive analysis of the orthometric heights' annual maximum fluvial levels records". Revise structure. 20) Page 6, lines 29-30: "It was assumed" or "we assumed" instead of "Were assumed that if ..." 21) Page 7, line 15: move the verb "Was performed" at the end of the sentence. 22) Page 8, line 25: "...shows" instead of "This return period shown"... 23) In the whole manuscript, I suggest just using "flood hazard", instead of "flood geohazard".
* * *

---

## Author Comment (AC1) · 1 Nov 2018

Dear Sir,

I have received your 'Referee Report' and, on behalf of the co-authors, I would like to thank you very much for your tireless effort in reviewing the manuscript and for your valuable comments, which will certainly improve our manuscript. We value the comments received greatly, as they have pointed out a number of issues to be addressed. We have answered all the comments of the reviewers. Answers are attached to this

letter. Along with the answers we are explaining all the changes we have done. We did modifications to the initial manuscript based on the suggestions of the reviewers. We hope that the editor will find the paper suitable for publication. Thank you very much for your kind consideration of this resubmitted version of our manuscript.

Sincerely yours,

Paulo Victor do Nascimento Araújo (On behalf of the authors of the manuscript)

ANSWERS TO REVIEWERS COMMENTS

MINOR CORRECTIONS: All minor corrections indicated by reviewers have been AC-CEPTED by the authors and ALREADY MODIFIED.

GENERAL COMMENTS: 1) [Editor]: Authors has used flood simulation model, how-ever the description of model to prepare the flood hazard map is missing. [ACCEPTED AND MODIFIED] [Authors]: The flood simulation model is based on the fill of the DEM calibrated at the river level orthometric heights, linked to a common geodetic reference system [included in text, page 5, line 1]. A flowchart was drawn up and included in the new text (figure 4).

2) [Editor]: Authors have utilized SRTM DEM 1 arc data set. Although DEM of a study area has not been provided in form of figure or map. It is important to include. [AC-CEPTED AND MODIFIED] [Authors]: A figure was drawn up and included in the new text (figure 7).

3) [Editor]: How the SRTM DEM is used to prepare the flood hazard maps? It is im-portant to describe. [ACCEPTED AND MODIFIED] [Authors]: Digital Elevation Model (DEM) is a set of digital data describing elevation values of Earth ground surface (or any other surface) which contains additional information about the character of this sur-face and interpolation algorithm, which is the best for approximation (modelling) of the real topography [According to (Szypuła, 2017)]. A DEM is a complete representation of a land surface which means that heights are available at each point in the area of

interest [According to (Hengl and Evans, 2009)]. In this study, was taken as the topographic model the SRTM DEM. However, it was submitted to the calibration process for adjustment to the local reference geodetic system [included in text, page 6, line 3]. For better understanding, a flowchart was drawn up and included in new text (figure 4).

4) [Editor]: What is Geoprocessing? Which geoprocessing technique has been used by authors to prepare a flood hazard maps? Explain. [ACCEPTED AND MODIFIED] [Authors]: Geoprocessing is a set of techniques based on the study of spatially distributed information in order to describe the characteristics of the phenomenon under investigation at the whole area of interest [According to (Costa and Lourenço, 2011)] [included in text, page 2, line 12]. Digital Image Processing (DIP), digital cartography and Geographic Information Systems (GIS) are undertakings of geoprocessing. In this work the use of geoprocessing techniques was extremely important to reach the results. And we use following techniques: [included in text, page 8, line 5] ïČij Geographic Information System (GIS): Technique most used practically throughout the work. All data was served to implement a robust GIS; ïČij Digital Cartography: During the elaboration of the maps; ïČij Digital Image Processing (PDI): Applied technique to improve the visualization of the historical flooding in the CBERS-4/MUX satellite scene; ïČij Precision Geodesy: During the obtaining of the points of land controls and linkage of the river level to the Brazilian Geodetic System; ïČij Geostatistical: During the evaluation and calibration of the Digital Elevation Model. Reference: Costa, S. B. and Lourenço, R. W.: Geoprocessing applied to the assessment of environmental noise: a case study in the city of Sorocaba, São Paulo, Brazil. Environmental Monitoring and Assessment, 172, 329–337, doi: <https://doi.org/10.1007/s10661-010-1337-3>, 2011.

5) [Editor]: Any rational of selections of flood hazard classes in Table 1 and Fig 6, if so, please explain. [ACCEPTED AND MODIFIED] [Authors]: To determine the classes of flood hazard mapping, a descriptive analysis of the orthometric heights time serie (annual maximum fluvial levels records) of Uruguay River was performed (minimum, maximum, quartile and percentile). The determination of the classes was closely linked

to the probability of occurrence of height annual maximum fluvial of Uruguay river [included in text, page 7, line 12]. At this stage, 5 classes of flood hazard were determined as described in table 1.

6) [Editor]: Table 2, Kendall's tau statistic 0.167, what its correlation with flood hazard mapping? It is important to explain significance of Mann Kendall test in flood mapping. [ACCEPTED AND MODIFIED] [Authors]: The nonparametric Mann-Kendall test, also known as Kendall's $\tau$au test or the Mann-Kendall trend test, is widely used to evaluate trends in time series. In recent years with growing concerns over environmental degradation and about the implications of green-house gases on the environment, researchers and practitioners have frequently applied the non-parametric Mann-Kendall test to detect trend in recorded hydrologic time series such as water quality, streamflow, and precipitation time series (Yue and Wang, 2004). Although it has no influence on the flood geohazard mapping; the Mann-Kendall test was applied to investigate if the elevation of the Uruguay river is showing any upward or downward trend [include in text, page 5, line 24].
* * *
[Figure]

**Fig. 1.** Figure 4: Flowchart of the proposed approach for delimitation of flood geohazard mapping.

[Figure]

**Fig. 2.** Figure 7: Digital Elevation Model (DEM): a) SRTM Original; b) SRTM Calibrated.

---

## Referee Comment (RC2) · Anonymous Referee #2 · 7 Nov 2018

Manuscript ID: nhess-2018-212

Title: Delimitation of Flood Areas Based on Calibrated DEM and Geoprocessing: Case Study on Uruguay River, Itaqui City, Southern Brazil

Authors: Paulo Victor. N. Araújo, Venerando E. Amaro, Robert M. Silva, Alexandre B. Lopes

OVERALL EVALUATION The manuscript focuses on the mapping of flood prone areas by means of calibrated DEM, historical series of streamflow records and geoprocessing techniques in Southern Brazil, in particular close to the city of Itaqui in the Uruguay River basin. After having calibrated the DEM using Ground Control Points, the floodable areas are mapped in relation to the results of a statistic analysis on the annual maximum level records of the Uruguay River, that identify 5 hazard classes. Maps resulting from of this procedure are then compared with the extent of the flooded area in two historical floods, showing a good similarity. Looking in a comprehensive way at the whole study, I would say that the core idea of the study is interesting for research purposes and fit perfectly in the context of the journal. Although this consideration, in my opinion the manuscript needs to be deeply revised in some points, in order to be published in NHESS. Authors can find my comments below, I hope the authors will find them useful.

General comments: I would give much more importance to the core of the manuscript, i.e. the mapping of floodable areas. Calibration and geoprocessing procedures are also important, but I would deepen and detail the description of the methods used to map the hazard of the study area, for the sake of reproducibility of the study, going into the details in a clear way and neglecting information, which are not connected with the analysis. For example, the current version of the manuscript does not allow readers to understand why authors relate levels in the river with water depth in the floodable areas: has the river no embankments at all? This might be clarified. Is it realistic that all areas with the same elevation in the study zone are affected by the same hazard, even if their distance from the river is some kilometres greater? Are there no obstacles or topographic discontinuities that can influence flooding dynamics? In my opinion, these are aspects that might be discussed in the paper, in order to improve the robustness of the methodology. In addition, I would better clarify the reason behind the choice of the relationship between the river level statistics and the hazard classes: for example, are there other literature studies that justify this selection? Furthermore, it is not clear to me what the "simulated flood altimetric quota" mentioned at the beginning of Section 4 is: probably, the word "simulated" is misleading, and it only identifies areas, which are below a certain terrain elevation? Considering the introduction, I would suggest

[Figure]

to detail the aim of the study and why authors use the methodology they describe, giving an overview of the literature background of the topic: other procedure used for mapping flood hazard (1D-2D hydrodynamic models, other DEM-based method, just to cite some of them), their advantages and disadvantages also focusing on the specific case study, in order to justify the developed procedure. The study area could be shortened a bit, neglecting information that are not very useful for the focus of the paper. I think the extension of the study area is wrong, because it seems to be much greater than the total area of the Uruguay River basin and it doesn't match with figure 1. According to the description of the procedure in Section 4.2.1, I would expect the independent variable (GCPs) on the x-axis and SRTM data on the y-axis, both axes ranging from the same minimum to the same maximum. In my opinion, a figure showing a comparison between original and calibrated DEM (with the same colour scale range) would be useful to better understand the improvements coming from the DEM's calibration. As last comment, I find the validation part of the paper, i.e. the comparison of the study results with the historical flood area extensions, too short and superficial, while it should represent one of the most important part of the manuscript. A "visual comparison" (see also the abstract) without numeric and statistics results is, in my opinion, not suitable for a scientific research paper and cannot be used to draw reliable conclusions about the good performance of the methodology.

Minor comments: I would recommend revising the language: there are a lot of misspellings and grammatical errors, and together with the complexity and the ambiguity of some sentences, they make the manuscript sometimes difficult to understand. As general correction, make sure that every sentence has a subject and put the verb in the correct position, not at the beginning. Furthermore, make sure of the temporal coherence of verbs, because now some sentences have the present, some other have the past. Abstract, p. 1 line 2: historical instead of historic Abstract, p. 1 l. 16: what does "submitted the statistical analysis" mean? Introduction, p. 2 line 13: what is the "sound judgements of the modeller"? Study area, p. 2 line 26 and others: pay attention to the units, please write the km2 with the superscript function. Study area, p. 3 line 11: I can

see only ten sub-basins, although it is written that they are eleven. Study area, p. 3 line 14: The study area comprises the urban area of Itaqui city AND is located... Study area, p. 3 line 16: official instead of oficial Section 3, p. 4 l. 5: relevant problem to the local population, only... (without "and") Section 3, p. 4 line 6: risks instead of riscks Section 4.1: the reference to Fig 4 is missing Section 4.2.1, p. 6 line 5-6: what does "in good conservation" mean?

---

## Editor Comment (EC2) · D. Patel (Editor) · 17 Nov 2018

Dear Reviewer(s),

Thank you for reviewing the manuscript nhess-2018-212, entitled " Delimitation of Flood Areas Based on Calibrated DEM and Geoprocessing: Case Study on Uruguay River, Itaqui City, Southern Brazil" for NHESS.

We greatly appreciate the voluntary contribution that each reviewer gives to the Journal. We hope that we may continue to seek your assistance with the refereeing process

for NHESS, and hope also to receive your own research papers that are appropriate to our aims and scope.

Yours Sincerely, Dr. Dhruvesh Patel Guest Editor, NHESS dhruvesh.patel@pdpu.ac.in
* * *

---

## Editor Comment (EC4) · D. Patel (Editor) · 17 Nov 2018

Dear Paulo Victor N. Araújo et al.,

We have received the reports from our reviewers on your manuscript, " Delimitation of Flood Areas Based on Calibrated DEM and Geoprocessing: Case Study on Uruguay River, Itaqui City, Southern Brazil ", submitted to NHESS.

Based on the advice received, I have decided that your manuscript can be accepted for publication after you have carried out the rigorous corrections as suggested by the

reviewer(s) and attached in revised manuscript, Therefore the decision that we arrived for this article was: Publish Subject to major revision including technical and language corrections.

Please make sure to submit your editable manuscript files (i. e. Word, PDF)."

Please submit your revised manuscript online by using the Editorial Manager system.

I am looking forward to receiving your revised manuscript in a one week period.

With kind regards,

Dr. Dhruvesh Patel Guest Editor, NHESS dhruvesh.patel@pdpu.ac.in
* * *

---

## Author Comment (AC3) · 17 Dec 2018

Dear Editor and the reviewer,

We do appreciate your constructive, thoughtful, careful, and helpful comments and suggestions. After careful discussions and analyses, we finished the preparation of responses to you. If there are any new comments or suggestions, please let us know.

Best Regards,

Paulo Victor N Araújo and the coauthors

**Response to General Comment:**
1) [Ph.D. Samela]: The introduction is focused almost exclusively on the importance and the role of flood hazard maps. The background of the research has not been delineated. Traditional procedures for flood hazard studies, or alternative methods, and the problems/limits related to both of them are not mentioned. Therefore, it is not clear what is the gap or issue that the proposed research aim to address? In few words, the aims are not clearly defined. Also, data, methods, models, performance measures should be illustrated in more detail.
[Authors´s answer]: [ACCEPTED and MODIFIED INTRODUCTION]

2) [Ph.D. Samela]: The extent of the study area is not clear. Section "Study area" reports that the full Uruguay basin has a total area of 385,000 km2. Then, it is also reported that the study area corresponds to Ibicui sub-basin, the largest Uruguay river sub-basin, and that this study area has a territorial area of 3,406,606 km2. How can this sub-basin have a drainage area larger than the full hydrographic basin?
[Authors´s answer]: We apologize for the punctuation and correct it in the text. MODIFIED FOR "… and that this study area has a territorial area of 34,066.06 Km2 …".

3) [Ph.D. Samela]: A SRTM-DEM has been calibrated using ground control points (GCPs) of high vertical accuracy. Can you provide a quantitative indication about GCPs vertical accuracy?
[Authors´s answer]: [ACCEPTED and MODIFIED]: "The database with 697 control points from Federal University of Pampa (UNIPAMPA) were acquired with GNSS receivers on the field through high-accuracy post-processed kinematic (PPK) mode and linked to SGB (Silva et al., 2017). These high-accuracy control points had the mean of the altimetric error the value of 0.006 ± 0.0007 meters" [page 6, line 26].

4) [Ph.D. Samela]: In carrying out the visual comparison in Section 4, please, explain more clearly what do you mean with "simulated flood altimetric quota". More details as regards the simulation need to be provided.
[Authors´s answer]: MODIFIED FOR: "Finally, a visual comparison between a modelled flood versus a DIP from flood area satellite image was performed, which both were registered concomitantly on the same day in study region" [page 4, line 28].

5) [Ph.D. Samela]: Section "4.2.1 Digital Elevation Model (DEM) calibration" specifies that in performing the linear regression, GCPs values have been used as independent variable and SRTM data as dependent variable. The independent variable is usually a measurement you are not manipulating in your experiment, and conventionally it is on the x axis. Instead Figure 5 puts SRTM values on the x-axis. Can you clarify Figure 5, the linear function y=0.7031x+13.913 you derived, and how did you use it?
[Authors´s answer]: We apologize for the mistake in writing the text, but in fact the independent variable was SRTM values and dependent variable was GCPs values. Common procedure found in the literature for DEM calibration (e.g., Gorokhovich and Voustianiouk, 2006; Du et al., 2012; Forkuor and Maathuis, 2012).

[TEXT MODIFIED FOR]: "This dataset was submitted to Linear Regression analysis, with ground control point values as dependent variable and SRTM data as independent variable. Common procedure found in the literature for DEM calibration (e.g., Gorokhovich and Voustianiouk, 2006; Forkuor and Maathuis, 2012)." [page 5, line 6].

**Reference**

Forkuor G. and Maathuis, B.: Comparison of SRTM and ASTER Derived Digital Elevation Models over Two Regions in Ghana – Implications for Hydrological and Environmental Modeling, in: Studies on Environmental and Applied Geomorphology, edited by: Piacentini, M. and Miccadei E., IntechOpen, https://doi.org/10.5772/28951, 2012.

Gorokhovich Y. and Voustianiouk A.: Accuracy assessment of the processed SRTM-based elevation data by CGIAR using field data from USA and Thailand and its relation to the terrain characteristics, Remote Sensing of Environment, 104, 409-415, https://doi.org/10.1016/j.rse.2006.05.012, 2006.

6) [Ph.D. Samela]: As far as I understand, the function obtained in Linear Regression has been used to predict the dependent variable values (the DEM values) as a function of the GCPs. Then the original SRTM DEM and the DEM adjusted with GCPs have been compared and RMSE has been evaluated. More interesting, in my opinion, would be to make a statistical comparison between the "new" adjusted DEM values against GCPs different from the ones used for the calibration, in order to validate the improvement in accuracy produced by this procedure.

[Authors´s answer]: We understand the concern, however for this study, we defined use of 100% of the control points for calibration and evaluation, as found in the literature (e.g., Araújo and Amaral, 2016).

**Reference**

Araújo, P.V.N. and Amaral, R.F.: Mapping of coral reefs in the continental shelf of Brazilian Northeast through remote sensing, Journal of Integrated Coastal Zone Management, 16, 5-20, http://dx.doi.org/10.5894/rgci629, 2016.

7) [Ph.D. Samela]: As regards the comparison showed in Figure 8 between the results of the proposed approach and CBERS-4/MUX satellite image for 12 June 2017, I suggest complementing this visual comparison with some statistics and performance measures. I believe this validation will improve the manuscript and the reliability of the proposed method.

[Authors´s answer]: It would be very interesting this strategy of validation, however to materialize it we faced the spectral limitation of the MUX sensor of CBERS-4 satellite. Due to the existence of trees with high crowns, in sectors affected by the flood, these would be masked in the PDI process. In our strategy, we opted to perform only comparative visual analysis. A future solution would be the use of other sensors, which are not currently available to authors.

**Response to Minor comments:**

1) Line 2, Abstract: replace "historic" with "historical". [ACCEPTED and MODIFIED]

2) Line 15, Abstract: instead of "fluviometric temporal series records" I suggest "temporal series of streamflow records". [ACCEPTED and MODIFIED FOR]: "…temporal series of maximum annual level records of Uruguay river…"

3) Line 15-16, Abstract: Check subject-verb agreement in "The annual maximum. . .were linked to. . .". [ACCEPTED and MODIFIED FOR]: The temporal series of maximum annual level records of Uruguay River, for years of 1942 to 2017, to Brazilian Geodetic System were linked using geometric levelling and submitted the descriptive statistical analysis and probability.

4) Line 16, Abstract: "submitted the statistical analysis". Unclear. [ACCEPTED and MODIFIED]: "submitted the descriptive statistical analysis and probability".

5) Line 18-19, Abstract: "Using the temporal series statistical analysis results, was assessed the spatialisation of flood hazard classes on the calibrated DEM and validated". Please, rephrase and move the verb "was assessed" after its subject "the spatialisation of flood hazard classes on the calibrated DEM". [ACCEPTED and MODIFIED]

6) Line 23, Abstract: instead of "Were determinate 5 classes of flood hazards", move the verb at the end of the sentence and correct it in "were determined". [ACCEPTED and MODIFIED]

7) Line 28, Introduction: check subject-verb agreement: "causes" instead of "cause". [ACCEPTED and MODIFIED]

8) Page 2, lines 3-7, Introduction: "These geohazards can be prevented and reduced by providing reliable information to the public about the flood hazard through flood inundation maps (Alaghmand et al., 2010; Demir, 2015). Information about the flood's extension is extremely important to evaluate the hazard of flood-prone areas and to help the rescue operations during these events (Cook and Merwade, 2009). Flood hazard mapping is one of the tools used to help communities avoid or mitigate such losses and damages (Arrighi et al., 2013; Savage et al., 2014; Speckhann et al., 2017). Flood hazards maps need therefore to be created as they provide a basis for the development of flood risk management plans". 9) Please, rephrase and avoid repetition of the same concept.
[ACCEPTED and MODIFIED]: These hazards can be prevented and reduced by providing reliable information to the public about the flood hazard through flood inundation maps (Alaghmand et al., 2010; Demir, 2015). This information can, for example, assist urban management or even to help the rescue operations during these events (Cook and Merwade, 2009). Thus, helping the communities directly to avoid or mitigate such future losses and damages (Arrighi et al., 2013; Savage et al., 2014; Speckhann et al., 2017). Flood hazards maps need therefore to be created as they provide a basis for the development of flood risk management plans".

10) Page 2, lines 15-16, Introduction: use "high vertical accuracy" in "topographic data. . . must possess vertical highly accuracy altimetric". [ACCEPTED and MODIFIED]

11) Page 3, line 16, Study area: "official" instead of "oficial". [ACCEPTED and MODIFIED]

12) Page 3, line 15, Study area: correct "rive" with "river". [ACCEPTED and MODIFIED]

13) Page 4, Line 4, Section "3 Previous studies in Itaqui city on flooding": Check subject-verb agreement and grammar in "The flooding process of Uruguay River in Itaqui city are a natural phenomenon that afflicts the riverside population for decades". [ACCEPTED and MODIFIED]

14) Page 4, Line 6, Section "3 Previous studies in Itaqui city on flooding": "risks" instead of "riscks". [ACCEPTED and MODIFIED]

15) Page 4, Line 15: "fulfill" instead of "fulfil". [ACCEPTED and MODIFIED]

16) Page 4, Lines 15,16: "priming in the use on the high elevation accuracy of altimetric and fluviometric data to the modelling of flood geohazard mapping". Revise English. [MODIFIED FOR]: "…priming in the methodological application of use of high accuracy altimetric data to the modelling of flood hazard mapping…".

17) Page 4, Lines 20-22: "was submitted the statistical analyses". Unclear. [ACCEPTED and MODIFIED]

18) Page 5, line 26: "Was considered as GCPs only the orthometric altimetry points acquired from high accuracy Geodesy which data were based in Global Navigation Satellite System (GNSS)". Revise English and subject-verb agreement. [ACCEPTED and MODIFIED FOR]: Was considered as GCPs the orthometric altimetry points acquired from high accuracy Geodesy.

19) Page 6, lines 28-29: "temporal series descriptive analysis of the orthometric heights' annual maximum fluvial levels records". Revise structure. [ACCEPTED and MODIFIED FOR]: "To determine the classes of flood hazard mapping, a descriptive analysis of the orthometric heights temporal series (annual maximum fluvial levels records) of Uruguay River was performed (minimum, maximum, quartile and percentile).".

20) Page 6, lines 29-30: "It was assumed" or "we assumed" instead of "Were assumed that if . . ." [ACCEPTED and MODIFIED]

21) Page 7, line 15: move the verb "Was performed" at the end of the sentence. [ACCEPTED and MODIFIED]

22) Page 8, line 25: ". . .shows" instead of "This return period shown". [ACCEPTED and MODIFIED]

23) In the whole manuscript, I suggest just using "flood hazard", instead of "flood geohazard". [ACCEPTED and MODIFIED]

---

## Author Comment (AC4) · 17 Dec 2018

Dear Editor and the reviewer,

We do appreciate your constructive, thoughtful, careful, and helpful comments and suggestions. After careful discussions and analyses, we finished the preparation of responses to you. If there are any new comments or suggestions, please let us know.

Best Regards,

Paulo Victor N Araújo and the coauthors

**Response to General comments:**

1) [Anonymous Referee #2]: The current version of the manuscript does not allow readers to understand why authors relate levels in the river with water depth in the floodable areas: has the river no embankments at all? This might be clarified.

[Authors´s answer]: Large part of the city of Itaqui, it is in flood plain area of the Uruguay river, as many of the riverside cities of the study area (Saueressig, 2012). The floods in these regions are intimately linked to the rise of the river level. Therefore, in the case under study, the river-level altimetry is the main driver of the flood hazard [included in text, page 10, line 14].

**Reference**

Saueressig, S. R.: Zoneamento das áreas de risco a inundação da área urbana de Itaqui-RS, M.S. Dissertation, Federal University of Santa Maria, Santa Maria-RS, Brazil, 101 pp., http://repositorio.ufsm.br/handle/1/9362, 2012.

2) [Anonymous Referee #2]: Is it realistic that all areas with the same elevation in the study zone are affected by the same hazard, even if their distance from the river is some kilometres greater? Are there no obstacles or topographic discontinuities that can influence flooding dynamics?

[Authors´s answer]: Not is it realistic that all areas with the same elevation in the study zone are affected by the same hazard. "In a few areas there was topographic discontinuity. When these altimetric class discontinuities occurred, the sector in focus was considered in the upper elevation altimetric class" [included in text, page 7, line 21].

3) [Anonymous Referee #2]: I would better clarify the reason behind the choice of the relationship between the river level statistics and the hazard classes: for example, are there other literature studies that justify this selection?

[Authors´s answer]: To determine the classes of flood hazard mapping, a descriptive analysis of the orthometric heights time series (annual maximum fluvial levels records) of Uruguay River was performed (minimum, maximum, quartile and percentile).

The determination of the classes was closely linked to the probability of occurrence of height annual maximum fluvial of Uruguay river [included in text, page 7, line 12]. At this stage, 5 classes of flood hazard were determined as described in table 1.

Table 1: Classes of flood hazard for mapping.

| Classes | Altimetric quota river used as indicator |
|---|---|
| Extremely high flood hazard | < Median |
| High flood hazard | ≥ Median and < 3rd quartile |
| Moderate flood hazard | ≥ 3rd quartile and < 95% |
| Low flood hazard | ≥ 95% and < Maximum quota |
| Non-floodable | > Maximum quota |

4) [Anonymous Referee #2]: It is not clear to me what the "simulated flood altimetric quota" mentioned at the beginning of Section 4 is: probably, the word "simulated" is misleading, and it only identifies areas, which are below a certain terrain elevation?

[Authors´s answer]: [MODIFIED FOR]: "Finally, a visual comparison between a modelled flood versus a DIP from flood area satellite image was performed, which both were registered concomitantly on the same day in study region".

5) [Anonymous Referee #2]: Considering the introduction, I would suggest to detail the aim of the study and why authors use the methodology they describe, giving an overview of the literature background of the topic: other procedure used for mapping flood hazard (1D-2D hydrodynamic models, other DEM-based method, just to cite some of them), their advantages and disadvantages also focusing on the specific case study, in order to justify the developed procedure.

[Authors´s answer]: [ACCEPTED and MODIFIED INTRODUCTION]

6) [Anonymous Referee #2]: The study area could be shortened a bit, neglecting information that are not very useful for the focus of the paper.

[Authors´s answer]: We find pertinent the permanence of the text of the study area, since, rarely, the articles found in the international literature reporting the aspects of the basin under study are rare.

7) [Anonymous Referee #2]: I think the extension of the study area is wrong, because it seems to be much greater than the total area of the Uruguay River basin and it doesn't match with figure 1.

[Authors´s answer]: We apologize for the punctuation and correct it in the text. MODIFIED FOR "… and that this study area has a territorial area of 34,066.06 km$^2$ …".

8) [Anonymous Referee #2]: According to the description of the procedure in Section 4.2.1, I would expect the independent variable (GCPs) on the x-axis and SRTM data on the y-axis, both axes ranging from the same minimum to the same maximum.

[Authors´s answer]: We apologize for the mistake in writing the text, but in fact the independent variable was SRTM values and dependent variable was GCPs values. Common procedure found in the literature for DEM calibration (e.g., Gorokhovich and Voustianiouk, 2006; Du et al., 2012; Forkuor and Maathuis, 2012).

[TEXT MODIFIED FOR]: "This dataset was submitted to Linear Regression analysis, with ground control point values as dependent variable and SRTM data as independent variable. Common procedure found in the literature for DEM calibration (e.g., Gorokhovich and Voustianiouk, 2006; Forkuor and Maathuis, 2012)." [page 5, line 6].

**Reference**

Forkuor G. and Maathuis, B.: Comparison of SRTM and ASTER Derived Digital Elevation Models over Two Regions in Ghana – Implications for Hydrological and Environmental Modeling, in: Studies on Environmental and Applied Geomorphology, edited by: Piacentini, M. and Miccadei E., IntechOpen, https://doi.org/10.5772/28951, 2012.

Gorokhovich Y. and Voustianiouk A.: Accuracy assessment of the processed SRTM-based elevation data by CGIAR using field data from USA and Thailand and its relation to the terrain characteristics, Remote Sensing of Environment, 104, 409-415, https://doi.org/10.1016/j.rse.2006.05.012, 2006.

9) [Anonymous Referee #2]: In my opinion, a figure showing a comparison between original and calibrated DEM (with the same colour scale range) would be useful to better understand the improvements coming from the DEM's calibration.

[Authors´s answer]: A figure was drawn up and included in the new text (figure 7).

[Figure]

Figure 7: Digital Elevation Model (DEM): a) SRTM Original; b) SRTM Calibrated.

10) [Anonymous Referee #2]: I find the validation part of the paper, i.e. the comparison of the study results with the historical flood area extensions, too short and superficial, while it should represent one of the most important part of the manuscript. A "visual comparison" (see also the abstract) without numeric and statistics results is, in my opinion, not suitable for a scientific research paper and cannot be used to draw reliable conclusions about the good performance of the methodology.

[Authors´s answer]: It would be very interesting this strategy of validation, however to materialize it we faced the spectral limitation of the MUX sensor of CBERS-4 satellite. Due to the existence of trees with high crowns, in sectors affected by the flood, these would be masked in the PDI process. In our strategy, we opted to perform only comparative visual analysis. A future solution would be the use of other sensors, which are not currently available to authors.

**Response to Minor comments:**

11) [Anonymous Referee #2]: Abstract, p. 1 line 2: historical instead of historic.

[Authors´s answer]: [ACCEPTED AND CORRECTED].

12) [Anonymous Referee #2]: Abstract, p. 1 l. 16: what does "submitted the statistical analysis" mean?

[Authors´s answer]: [ACCEPTED and MODIFIED]: "submitted the descriptive statistical analysis and probability".

13) [Anonymous Referee #2]: Introduction, p. 2 line 13: what is the "sound judgements of the modeller"?
[Authors´s answer]: Several factors, such as: topographic discontinuity.

14) [Anonymous Referee #2]: Study area, p. 2 line 26 and others: pay attention to the units, please write the km2 with the superscript function.
[Authors´s answer]: [ACCEPTED AND CORRECTED].

15) [Anonymous Referee #2]: Study area, p. 3 line 11: I can see only ten sub-basins, although it is written that they are eleven.
 [Authors´s answer]: Not, the text is correct. "In Brazil, the Uruguay hydrographic region is composed by eleven hydrographic sub-basins: (1) Apuae–Inhandava, (2) Passo Fundo, (3) Turvo–Santa Rosa–Santo Cristo, (4) Piratinim, (5) Ibicui, (6) Quarai, (7) Santa Maria, (8) Negro, (9) Ijui, (10) Varzea and (11) Butui–Icamaqua". In new text, we add the number before the name, to guide the reader.

16) [Anonymous Referee #2]: Study area, p. 3 line 14: The study area comprises the urban area of Itaqui city AND is located.
[Authors´s answer]: [ACCEPTED AND CORRECTED].

17) [Anonymous Referee #2]:  Study area, p. 3 line 16: official instead of official.
[Authors´s answer]: [ACCEPTED AND CORRECTED].

18) [Anonymous Referee #2]: Section 3, p. 4 l. 5: relevant problem to the local population, only. . . (without "and").
[Authors´s answer]: [ACCEPTED AND CORRECTED].

19) [Anonymous Referee #2]: Section 3, p. 4 line 6: risks instead of riscks.
[Authors´s answer]: [ACCEPTED AND CORRECTED].

20) [Anonymous Referee #2]: Section 4.1: the reference to Fig 4 is missing.
[Authors´s answer]: All references to the figures were checked in the text. Everything is OK!

21) [Anonymous Referee #2]: Section 4.2.1, p. 6 line 5-6: what does "in good conservation" mean?
[Authors´s answer]: Existing still in the field and intact.

---

## Author Comment (AC5) · 17 Dec 2018

Dear PhD. Dhruvesh Patel,

We would like to thank you very much for yours tireless efforts in reviewing the manuscript and for yours valuables comments, which will certainly improve our manuscript. We greatly value the comments received, as they have pointed out several issues to be addressed. We have answered all the comments of the reviewers. We did modifications to the initial manuscript based on the suggestions of the review-

ers and it follows as attachment. We hope that the editor will find the paper suitable for publication. Thank you very much for your kind consideration of this resubmitted version.

Sincerely yours,

Paulo Victor do Nascimento Araújo (On behalf of the authors)

Please also note the supplement to this comment:
https://www.nat-hazards-earth-syst-sci-discuss.net/nhess-2018-212/nhess-2018-212-AC5-supplement.pdf

**Supplement:**

[revised manuscript text omitted]

---

## Editor Comment (EC5) · Patel (Editor) · 24 Dec 2018

Referees are requested to verify the revised version and provide suggestions in one week time
* * *